# An Accurate Doppler Parameters Calculation Method of Geosynchronous SAR Considering Real-Time Zero-Doppler Centroid Control

**Faguang Chang** [1] , **Chunrui Yu** [2], **Dexin Li** [1], **Yifei Ji** [1,*] **and Zhen Dong** [1]

1. College of Electronic Science and Technology, National University of Defense Technology, Changsha 410073, China; cfg@nudt.edu.cn (F.C.); lidexi@nudt.edu.cn (D.L.); dongzhen@nudt.edu.cn (Z.D.)
2. Beijing Institute of Tracking and Telecommunication Technology, Beijing 100094, China; ycrxc@163.com
* Correspondence: jiyifei@nudt.edu.cn

**Abstract:** The zero-Doppler centroid control in geosynchronous synthetic aperture radar (GEO SAR) is beneficial to reduce the imaging complexity (reduces range-azimuth coupling in received data), which can be realized by adjusting the radar line of sight (RLS). In order to maintain the zero-Doppler centroid throughout the whole orbit of the GEO SAR satellite, the RLS needs to be adjusted in real-time. Due to the ultra-long synthetic aperture time of GEO SAR, the RLS variation during the synthetic aperture time cannot be neglected. However, in the previous related papers, the real-time variation of RLS during the synthetic aperture time was not taken into account in the calculation of Doppler parameters, which are closely related to the RLS, resulting in inaccurate calculation of Doppler parameters. Considering this issue, an accurate Doppler model (the model of relative motion between satellite and ground target) of GEO SAR is proposed in this paper for the accurate calculation of Doppler parameters (Doppler centroid and Doppler bandwidth and other parameters). Finally, simulation experiments are designed to confirm the effectiveness and necessity of the proposed model. The results indicate that the RLS variation during the synthetic aperture time has a considerable effect on Doppler parameters performance of the GEO SAR, and refers to a more stable azimuth resolution performance (the resolution is kept near a relatively stable value at most positions of the elliptical orbit) compared with the case that does not consider the real-time zero-Doppler centroid control.

**Keywords:** geosynchronous synthetic aperture radar (GEO SAR); system design; zero-Doppler centroid control; radar line of sight (RLS) variation; Doppler parameters performance



## 1. Introduction

Since the concept of geosynchronous synthetic aperture radar (GEO SAR) was first proposed by K. Tomiyasu in 1978, it has been regarded as a potential SAR imaging mode to provide Earth observation with a wide swath and short revisit time (24 h) [1]. At present, the main research orbit scheme of GEO SAR includes near-zero inclined orbit and inclined orbit, and the near-zero inclined orbit is mainly studied by European scholars, while the inclined orbit is studied by American and Chinese scholars. Compared with the large antenna size and higher transmission power required by the highly inclined orbit, the theoretical research on the small inclined orbit scheme has made great progress 20 years ago [2,3]. Since 2000, with the rapid development of electronic technology, many studies of inclined GEO SAR have gained momentum again. In the inclined orbit scheme, the nadir-point trajectory of the high-inclination orbit is a large figure shape, which can provide a large observation area [4]. These unique advantages enable GEO SAR to not only provide surface coverage for approximately one-third of the globe but also greatly improve the response speed of emergencies in the interested area [5], which attracts increasing attention from engineers and scholars.

In space-borne SAR, the Doppler centroid is of particular importance in imaging processing [6,7]. In order to avoid the complex process of the Doppler centroid estimation and improve the imaging processing accuracy, a yaw steering method was proposed to obtain zero-Doppler centroid under the hypothesis of circular orbits for the first time [8,9]. However, the residual Doppler central frequency can be up to kilohertz when the satellite runs at the geosynchronous orbit with a non-negligible orbital eccentricity, due to the Earth rotation and elliptical orbit [10–12]. Reference [13] proposed an attitude steering method combining pitch and rotation to accurately reduce the residual Doppler center error when the look angle of radar is out of an expected range. However, this approach is difficult to implement in GEO SAR. According to the different attitude steering of pitch steering and yaw steering, two new methods of zero-Doppler centroid control called pitch-yaw steering and yaw-pitch steering were derived [10,14], but the yaw angle variation can reach several tens of degrees, which is impractical for the colossal platform of GEO SAR. The two-dimensional (2D) phase scanning described by look-down and squint angles is a good substitution for the attitude steering, which can be accomplished with a phased-array antenna. Moreover, we fully analyze the zero-Doppler centroid attitude steering control method in this paper. Since there are no geosynchronous satellites in orbit, it is not possible to obtain the measured data based on a specific hardware platform, but we can realize the accurate Doppler model through the real-time variation of the look-down angle and the squint angle throughout the orbital period.

With consideration of the zero-Doppler centroid control, Doppler parameters of the GEO SAR including the synthetic aperture time, Doppler bandwidth, and azimuth resolution can be calculated for the system design and demonstration. However, the real-time variation of the look-down and squint angles within the synthetic aperture time has never been considered in the calculation of Doppler parameters, which can bring about incorrect calculation results and lead to poor systematical designation [15]. This paper focuses on the study of the accurate Doppler model based on real-time zero-Doppler centroid control.

This paper is organized as follows. In Section 2, the real-time zero-Doppler control is introduced. In Section 3, the Doppler model considering the real-time variation of the look-down angle and squint angle within the synthetic aperture time is accurately derived. In Section 4, simulation experiments are designed to validate the effectiveness of the proposed model. Finally, Section 5 concludes the entire study, including a discussion on future research.

## 2. Real-Time Zero-Doppler Centroid Control

### 2.1. Accurate Attitude Steering Control

The geometry of the Earth-centered inertial (ECI) coordinate system (CS) and the mass-centered orbit (MCO) CS is illustrated in Figure 1, which refer to $xyz$ and $XYZ$, respectively. Furthermore, $\boldsymbol{R_s}$ and $\boldsymbol{R_t}$ represent the position vectors of the satellite and the target, respectively, $\boldsymbol{R}$ is denoted as $\boldsymbol{R_s} - \boldsymbol{R_t}$, whose absolute value is the slant range, and $S$ is the mass center of the satellite in the MCO CS [10]. Based on the derivations in [8,9], the calculation formula of the Doppler centroid can be presented as:

$$f_{dc} = -2\mathbf{P} \cdot \mathbf{Q} / \lambda \tag{1}$$

where $\mathbf{P}$ is a function of satellite position, $\mathbf{Q} = \boldsymbol{R}/R$ is the unit vector of the radar line of sight (RLS). Further derivation is carried out in the MCO CS, so vectors $\mathbf{P}$ and $\mathbf{Q}$ can be expressed as:

$$\begin{cases} \mathbf{P} = \left( R_s\left(\dot{\alpha} - \omega_e \cos\alpha_i\right), -R_s\omega_e \sin\alpha_i \cos\alpha, -\dot{R_s} \right) \\ \mathbf{Q} = (0, -\sin\gamma_0, -\cos\gamma_0) \end{cases} \tag{2}$$

where $\alpha_i$ is the orbital inclination angle, $\alpha$ is the argument of latitude, $\dot{\alpha}$ is the derivation of $\alpha$ with respect to time, $\omega_e$ is the Earth's rotational angular velocity, and $\gamma_0$ is the original look-down angle.

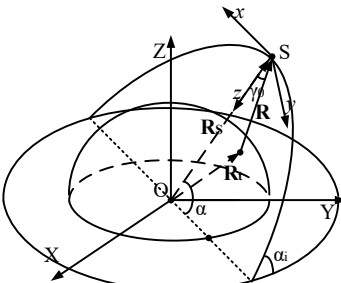

**Figure 1.** The geometry of ECI and MCO CS.

In order to realize the zero-Doppler centroid, the attitude steering is operated in different orders of pitch steering and yaw steering. The yaw angle and pitch angle needed to be adjusted by the two attitude steering control methods have different expressions. The pitch steering and yaw steering operators are expressed as follows:

$$\mathbf{M}_{\text{pitch}} = \begin{bmatrix} \cos\theta & 0 & -\sin\theta \\ 0 & 1 & 0 \\ \sin\theta & 0 & \cos\theta \end{bmatrix}, \mathbf{M}_{\text{yaw}} = \begin{bmatrix} \cos\varphi & \sin\varphi & 0 \\ -\sin\varphi & \cos\varphi & 0 \\ 0 & 0 & 1 \end{bmatrix} \tag{3}$$

where $\theta$ is the pitch angle and $\varphi$ is the yaw angle. Attitude steering is essentially adjusting the RLS by multiplying the vector $\mathbf{Q}$ by matrix operators. In this paper, we only consider the pitch-yaw steering attitude control; the zero-Doppler centroid controlling equation can be expressed as:

$$\mathbf{P} \cdot \left( \mathbf{M}_{\text{yaw}} \mathbf{M}_{\text{pitch}} \mathbf{Q} \right) = \mathbf{P} \cdot \mathbf{Q}_{\text{pitch-yaw}} = 0 \tag{4}$$

From Equation (4), the solution can be derived as:

$$\begin{cases} \varphi = \arctan\left( \dfrac{\omega_e \sin\alpha_i \cos\alpha}{\dot{\alpha} - \omega_e \cos\alpha_i} \right) \\ \theta = \arctan\left( -\dfrac{\dot{R}_s / R_s}{\left( \dot{\alpha} - \omega_e \cos\alpha_i \right) \cos\varphi + \omega_e \sin\alpha_i \cos\alpha \sin\varphi} \right) \end{cases} \tag{5}$$

Then, based on the parameters listed in Table 1, the variation curves of the yaw angle and pitch angle are drawn in Figure 2.

**Table 1.** System parameter.

| Parameter | Value | Parameter | Value |
|---|---|---|---|
| Semi-major axis | 42,164.17 km | Right ascension of ascending | 115° |
| Eccentricity | $1 \times 10^{-8}$ | Perigee | 270° |
| Orbital inclination | 60° | Incident angle | 20° |
| Carrier frequency | 1.25 GHz | Antenna size | 30 m × 30 m |
| Pulse duration | 2.5 μs | Chirp bandwidth | 30 MHz |

### 2.2. Attitude Steering Implement Method

As shown in Figure 2, the variation scope of the yaw angle can reach several tens of degrees during a satellite period, which is impractical for the colossal platform of GEO SAR. Therefore, an equivalent method called antenna phase centroid scanning has been proposed as a good substitute for the attitude steering, which can be perfectly accomplished with a phased-array antenna and intuitively described with look-down and squint angles.

In reality, the essence of platform rotation in attitude steering is for adjusting the RLS, and the unit vector of the adjusted RLS can be denoted as:

$$\mathbf{Q}' = (-\sin\gamma\cos\phi, -\sin\gamma\sin\phi, -\cos\gamma) \tag{6}$$

where $\gamma$ is the adjusted look-down angle and $\phi$ is the squint angle, and they are both defined in the MOC CS. By solving $\mathbf{Q}_{\text{pitch-yaw}} = \mathbf{Q}'$, we obtain the solution to antenna phase centroid scanning in Equation (7). The newly solved look-down angle and squint angle are related to the original look-down angle and orbital parameters.

$$\begin{cases} \gamma = \arccos(\cos\gamma_0\cos\theta) \\ \phi = \arccos\left(\dfrac{\sin\gamma_0\sin\varphi - \cos\gamma_0\cos\varphi\sin\theta}{\sqrt{(\sin\gamma_0\sin\varphi - \cos\gamma_0\cos\varphi\sin\theta)^2 + (\sin\gamma_0\cos\varphi + \cos\gamma_0\sin\varphi\sin\theta)^2}}\right) \end{cases} \tag{7}$$

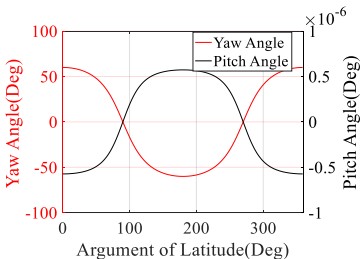

**Figure 2.** Pitch and yaw angles along the orbit.

Then, the variation curves of the squint angle and look-down angle in a satellite period are drawn in Figure 3.

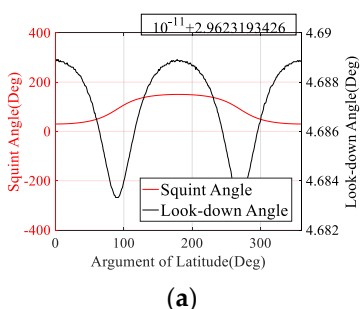

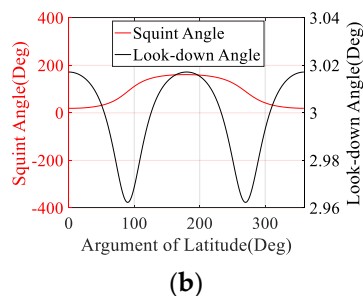

(a)           (b)

**Figure 3.** Squint and look-down angles along the orbit. (**a**) The eccentricity is $10^{-8}$; (**b**) The eccentricity is 0.01.

As shown in Figure 3, the squint angle varies in a larger scope compared with the look-down angle. Furthermore, with the increase of the eccentricity, the look-down angle also varies widely.

## 3. Accurate Doppler Model

In this section, a more accurate calculation method of Doppler parameters will be emphasized by considering the variation of the RLS within the synthetic aperture time compared with the formulas for calculating these parameters given in the previous related papers. In the application of zero-Doppler centroid control, the RLS should be continually adjusted along the orbit to obtain the zero-Doppler centroid, which is not only for the elliptical orbit but also for the near-circular orbit. Due to the ultra-long synthetic aperture time of GEO SAR, variations of the RLS including look-down angle and squint angle during the synthetic aperture time are considerable, which should be taken into account when Doppler parameters are analyzed.

The fluctuations of squint angle and look-down angle during the synthetic aperture time are analyzed. The pitch-yaw attitude steering control method is adopted and the argument of latitude is assumed to be $\alpha = 75°$. The RLS fluctuation within the synthetic aperture time is shown in Figure 4.

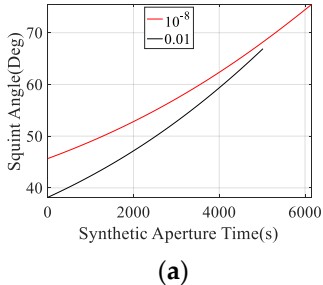 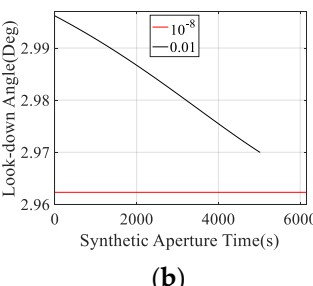

(**a**)                           (**b**)

**Figure 4.** Variations of the RLS during the synthetic aperture time. (**a**) The eccentricity is $10^{-8}$; (**b**) the eccentricity is 0.01.

As shown in Figure 4, the fluctuation scope of the squint angle is larger, which can reach tens of degrees during the synthetic aperture time. Furthermore, with the increase of the eccentricity, the variation of the look-down angle within the synthetic aperture time is greater. The look-down angle and the squint angle correspond to each other. Only when both of them have appropriate angles can the Doppler centroid be zero. It can be seen that even small fluctuation of the look-down angle plays a very significant role. Therefore, the variation of Doppler parameters caused by the real-time variation of the two must be fully considered within the synthetic aperture time. The following is an accurate analysis of Doppler parameters.

First, in order to illustrate the necessity of accurately calculating the Doppler parameters, the Doppler centroid results with or without considering the variation of the RLS during the synthetic aperture time are compared. The simulation results are shown in Figure 5.

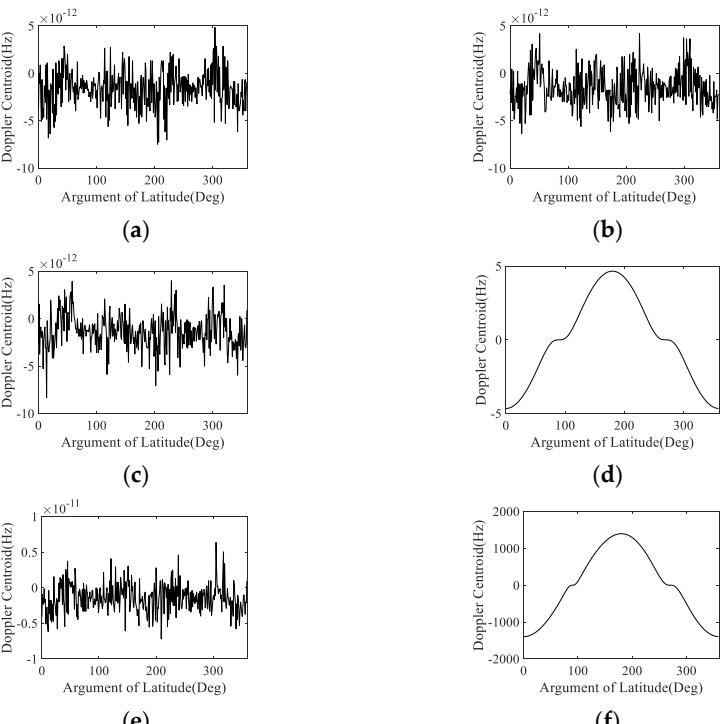

**Figure 5.** Doppler centroid along the orbit at different eccentricities. (**a,c,e**) consider the variation of the RLS, and the eccentricity is $10^{-8}$, 0.01, and 0.1, respectively. (**b,d,f**) without considering the variation of the RLS compared to (**a,c,e**).

As can be seen from Figure 5, without considering the variation of RLS within the synthetic aperture time, and when the eccentricity is small, the Doppler centroid residual can be ignored. However, when the eccentricity reaches 0.1, a large Doppler centroid residual will be generated, which will cause many problems in imaging processing. In contrast, when the variation of RLS is considered, the Doppler centroid residual is negligible even the eccentricity is large. Therefore, the real-time variation of the RLS must be considered in order to make the Doppler model more accurate. Then parameters of synthetic aperture time, azimuth bandwidth, and azimuth resolution are analyzed using the accurate Doppler model.

The geometry of the GEO SAR is completely different from that of the low-Earth-orbit SAR (LEO SAR). Therefore, the accuracy of the classical method of calculating synthetic aperture time is not sufficient for the GEO SAR system. A more accurate formula is derived to calculate synthetic aperture time ($T$). It can be accurately defined according to the 3 dB beam width ($\theta_{bw}$):

$$\begin{cases} \langle \boldsymbol{R_s}(t_1), \boldsymbol{R}(t_1) \rangle = \theta_{bw}/2 \\ \langle \boldsymbol{R_s}(t_2), \boldsymbol{R}(t_2) \rangle = -\theta_{bw}/2 \\ T = t_2 - t_1 \end{cases} \tag{8}$$

where $\langle \cdot \rangle$ is used to calculate the angle between two vectors, $t_1$ is the start instant when the target comes into the antenna beam coverage and $t_2$ is the end instant. In order to fully illustrate the relationship between the synthetic aperture time and various variables, we can use geometric knowledge to express it with mathematical expressions in Equation (9). The geometric model is shown in Figure 6.

$$\begin{cases} T_a = |t_2 - t_1| = \frac{R_c}{V_g} = \\[4pt] \dfrac{R_e \left[ \arcsin\left( \frac{|R_s(t_1)| + |R_s(t_2)|}{R_e} \sin\left( \frac{0.443\lambda}{L_a} \right) \right) - \frac{0.443\lambda}{L_a} \times 2 \right]}{V_{s-g}^2 + V_{e-g}^2 - 2V_{s-g} V_{e-g} \cos(k(\gamma,\phi)\alpha_i \cos(k(\gamma,\phi)\alpha))} \\[6pt] V_{s-g} = \sqrt{\frac{\mu}{a^3}} R_e \\[6pt] V_{e-g} = R_e w_e \cos\left[ \arcsin(\sin(\alpha_i)\sin(\alpha)) - \cos(k(\gamma,\phi)\alpha_i \cos(k(\gamma,\phi)\alpha)) \arcsin\left( \frac{|R_s(t)| \sin(\gamma)}{R_e} \right) - \gamma \right] \end{cases} \tag{9}$$

where $R_e$ is the radius of the Earth, $\theta$ is the angle between the satellite and the center of the Earth during synthetic aperture time, $R_c$ is the distance formed by the RLS trajectory on the Earth surface within synthetic aperture time, $\mu$ is constant of earth gravitation, $a$ is the semi-major axis, $V_g$ is the ground velocity, which is attributed to the movement of satellite $V_{s-g}$ and the rotation of earth $V_{e-g}$, $\lambda$ is the signal wavelength, $L_a$ is the antenna azimuth size, and $t$ is the azimuth time. $k(\gamma, \phi)$ is a coefficient introduced by $\gamma$ and $\phi$. It can be seen from Equation (9) that the synthetic aperture time is related to the orbital elements, wavelength, look-down angle, and squint angle. In addition, both the ground velocity and the two beam angles change during the synthetic aperture time, which is consistent with the results in Figure 4. According to Equations (8) and (9), the synthetic aperture time can not only be calculated accurately, but also can be intuitively analyzed for the convenience of system design.

Next, the azimuth bandwidth is analyzed, which is of great significance to the azimuth resolution. In future GEO SAR systems, the azimuth bandwidth should be calculated as accurately as possible to meet the processing accuracy requirements. The high order Taylor expansion slant range model is adopted in the GEO SAR systems [16,17]. Calculation of the azimuth frequency difference between the beginning and ending positions within the

synthetic aperture time is the basic idea of azimuth bandwidth $B_a$ calculation, as shown in Equation (10) [18].

$$\begin{cases} f(t) = -\frac{2}{\lambda} \cdot \frac{dR(t)}{dt} = -\frac{2}{\lambda} \cdot \sum_{n=1}^{N} n \cdot k_n \cdot t^{n-1} \\ B_a = f_{\max} - f_{\min} = \frac{2}{\lambda} \cdot \sum_{n=1}^{N} n \cdot k_n \cdot (t_2^{n-1} - t_1^{n-1}) \end{cases} \quad (10)$$

where $f$ is the instantaneous Doppler frequency, $R(t)$ is the variable slant range within the synthetic aperture time, and $k_n(n = 1, 2, \cdots, N)$ is the coefficients of the higher-order Taylor expansion slant range model. It can be seen from Equation (10) that the calculation of azimuth bandwidth is related to the synthetic aperture time. Therefore, when accurately calculating the azimuth bandwidth, it is necessary to consider the variation of the RLS within the synthetic aperture time.

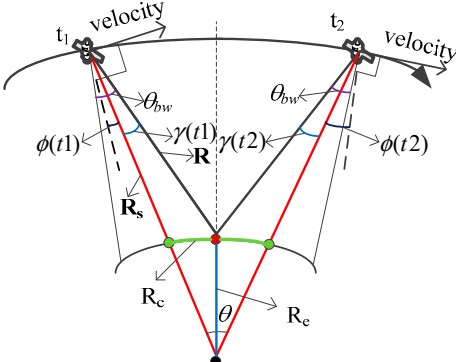

**Figure 6.** Geometric model of synthetic aperture time calculation.

Finally, another important Doppler parameter, azimuth resolution, is analyzed accurately. The choice of azimuth resolution depends on the requirements of radar application scenarios and must be considered in system design. The expression of azimuth resolution $\rho_a$ can be obtained based on the accurate azimuth bandwidth, as shown in Equation (11).

$$\rho_a = 0.886 \cdot \frac{V_g}{B_a} \quad (11)$$

The expression for the ground velocity $V_g$ has been given as Equation (9). It is easy to see from Equation (11) that the accurate calculation of the azimuth resolution is also related to the look-down angle and the squint angle, and their angle variation must be considered within the synthetic aperture time.

According to the above analysis, it can be concluded that the calculation of the three Doppler parameters, synthetic aperture time, azimuth bandwidth, and azimuth resolution, are directly or indirectly related to the look-down angle and the squint angle. Moreover, they play a vital role in the system design and SAR processing, therefore, we must accurately calculate them to meet the higher level of application requirements.

## 4. Simulations

Here, numerical experiments are carried out to validate the accuracy of the proposed Doppler model. We compare the results with or without considering the variation of RLS within the synthetic aperture time. The simulation of the synthetic aperture time, azimuth bandwidth, and azimuth resolution are respectively carried out. The parameters used are shown in Table 1.

First, we give the comparison results of synthetic aperture time in different look-down angles ($0°$, $1.5°$, and $3°$, and $0°$ is simply a reference and will not be used in the real system). The result is shown in Figure 7.

It can be seen from Figure 7 that the variation of RLS direction within the synthetic aperture time has a great influence on the calculation of synthetic aperture time. In some parts of the orbit, if the real-time variation of RLS is ignored, it will bring great error to SAR imaging. With the increase in the look-down angle, the synthetic aperture time becomes longer in general. Compared with the results which ignore the variation of RLS, the results calculated by the accurate Doppler model are mostly larger than the latter, except for some singularity positions.

Then, we simulate the azimuth bandwidth and give the comparison results whether the variation of RLS is considered within the synthetic aperture time. The comparison results are shown in Figure 8.

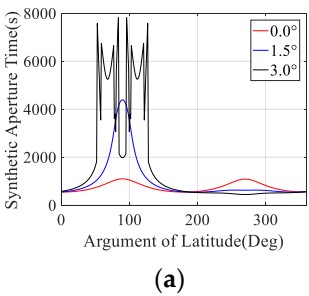 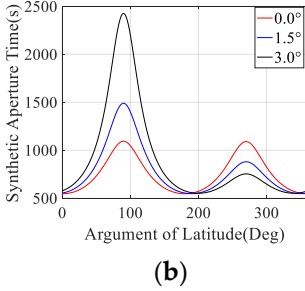

(**a**)    (**b**)

**Figure 7.** Synthetic aperture time along the orbit. (**a**) With the consideration of variations of the RLS; (**b**) without the consideration of variations of the RLS.

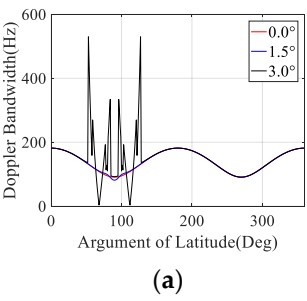 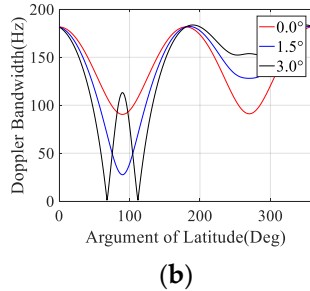

(**a**)    (**b**)

**Figure 8.** Doppler bandwidth along the orbit. (**a**) With the consideration of variations of the RLS; (**b**) without the consideration of variations of the RLS.

As can be seen from the comparison results in Figure 8, results vary greatly as to whether the variation of RLS is considered within the synthetic aperture time, especially at some singularity positions. If the pulse repetition frequency (PRF) is designed without considering the variation of RLS, it will cause Doppler ambiguity, which will bring a range of issues to the imaging.

Finally, the comparison results of azimuth resolution are given. The results are shown in Figure 9.

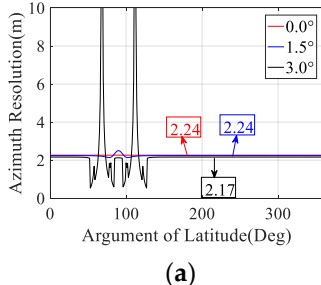 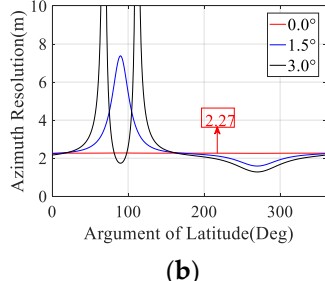

(**a**)    (**b**)

**Figure 9.** Azimuth resolution along the orbit. (**a**) With the consideration of variations of the RLS; (**b**) without the consideration of variations of the RLS.

It can be seen from Figure 9 that whether the variation of RLS is considered has a great influence on the calculation of the azimuth resolution, and the difference becomes more remarkable with the increase in the look-down angle. In addition, when the variation of RLS is considered, the constant azimuth resolution of the full aperture can be obtained except for some singularity positions.

For the figure-8-like orbit, the observation area can be controlled on the outer or inner side of the orbit by selecting the left-looking or right-looking. These singularities occur when the area is located on the inside and the initial look-down angle is small because the platform velocity is nearby the same as the target (derived from the rotation). In this case, zero-Doppler centroid can be abandoned, and squint mode imaging can be used to obtain stable Doppler performance, which can increase the flexibility.

## 5. Discussion

It is important to make the Doppler centroid frequency zero in GEO SAR. In this paper, the zero-Doppler centroid is realized by the pitch-yaw steering attitude control, and the concrete realization method in the GEO SAR system is given. In the actual GEO SAR system, the RLS is adjusted in real time. If the real-time variation is not considered in the calculation of system parameters, a large calculation error will be introduced, resulting in the system design not meeting the expectations. In order to improve the calculation accuracy, an accurate Doppler calculation model with full consideration of real-time variation of RLS is proposed. Finally, simulation experiments are conducted to compare the results of the two groups with or without considering the variation of RLS within the synthetic aperture time in different look-down angles. The results show that the difference between the two groups of results is more significant as the look-down angle increases, thus demonstrating the necessity of the proposed accurate Doppler model. The zero-Doppler is not always persistently adopted in GEO SAR. Squint mode is a quite common working strategy for wide coverage and can improve the flexibility of system design. However, the range-azimuth coupling is too severe to compensate. We will study the squint mode imaging algorithm in the future to improve the imaging accuracy.

**Author Contributions:** All authors have made substantial contributions to this work. F.C. and Y.J. formulated the theoretical framework. F.C. designed the simulations; F.C. carried out the simulation experiments; F.C., D.L. and Z.D. analyzed the simulated data; F.C. wrote the manuscript; D.L., C.Y. and Z.D. reviewed and edited the manuscript; Z.D. gave insightful and enlightening suggestions for this manuscript. All authors have read and agreed to the published version of the manuscript.

**Funding:** This research received no external funding.

**Institutional Review Board Statement:** Not applicable.

**Informed Consent Statement:** Not applicable.

**Data Availability Statement:** The data presented in this study are available on request from the corresponding author.

**Acknowledgments:** The authors would like to thank all those who gave valuable help and suggestions to this manuscript, which were essential to the outcome of this paper.

**Conflicts of Interest:** The authors declare no conflict of interest.

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
