# Peer review of "An Accurate Doppler Parameters Calculation Method of Geosynchronous SAR Considering Real-Time Zero-Doppler Centroid Control"

_remotesensing, doi:10.3390/rs13204061_

Round 1
Reviewer 1 Report
The paper deals with the unusual geometry of a geo synchronous SAR in the version to be adopted by China, namely with an inclination of 60 degrees, to be able to survey Northern countries. In that case, the figure of 8 nadir track and the relative slowness of the satellite motion wrt to the actual ground motion implicates a very long time (even hours) to be able to achieve a sufficient azimuth aperture. The innovation of the papers indeed only stems from the observation that during the very long aperture time (in real time in their parlance) work should be carried out to properly redirect the line of sight of the SAR system. Only with this redirection, a centered Doppler bandwidth, wide enough, can be obtained. To my opinion, the paper is correct and clearly understandable. Sure enough, the unique technical observation is that without an electronic beam steering that can adjust the Line of Sight of the beam in seconds, that geosynchronous SAR system could not operate well in all conditions and for many incident angles. To my opinion, these observations are relevant and should be made available to the readers. Hence, it is my opinion that the paper should be published. I was able to understand the concepts as written but still I would be favorable to accept the requests of the other reviewer who indeed noticed some imprecision that, however, still allows the informed reader to catch the concept. As I was able to read the observations of reviewer 2, in fact I agree with most of them. However, that reviewer does not deny the fact that the redirection of the beam in real time is mandatory: it is this the observation that makes the paper worth publication. Indeed the point is touchy, as it corrects for some excessive approximations made in previous publications on that theme, that are duly referred to in the paper.for a better appreciation of the possibilities of geosynchronous SAR it would be useful to add some references to the small inclination orbit case, as discussed in
Cheng Hu, Zhiyang Chen, Yuanhao Li, Xichao Dong, Stephen Hobbs,
Research progress on geosynchronous synthetic aperture radar, Fundamental Research, Volume 1, Issue 3, 2021, Pages 346-363,
ISSN 2667-3258, https://doi.org/10.1016/j.fmre.2021.04.008.
Monti Guarnieri, A., Rocca, F. Options for continuous radar Earth observations. Sci. China Inf. Sci. 60, 060301 (2017). https://doi.org/10.1007/s11432-016-9067-7
Reviewer 2 Report
In this paper, the authors proposed the Doppler model of GEO SAR for the accurate calculation of Doppler parameters. This model includes real-time variation of RLS (Radar Line of Sight). The following comments were addressed to the authors:
- It is necessary for the authors of this paper to compare their approach in solving Zero-Doppler Centroid Control problems with the approach presented in the paper: Yicheng Jiang, Bin Hu, Yun Zhang, Meng Lian, Zhuoqun Wang, "Study on Zero-Doppler Centroid Control for GEO SAR Ground Observation", International Journal of Antennas and Propagation, vol. 2014, Article ID 549269, 7 pages, 2014. https://doi.org/10.1155/2014/549269.
- The model proposed by the authors of this paper involves setting the RLS parameters in real time. Accordingly, it is necessary for the authors of this paper to provide information on the real-time performance of their model on a specific hardware platform that would be used in the actual implementation in the GEO SAR system.
Reviewer 3 Report
This is a paper with the title of “an accurate model of geosynchronous SAR considering real-time zero-Doppler centroid control”.
In this paper, authors have tried to investigate the effects of RLS variation on Doppler centroid for GEOSAR application!
Unfortunately, this article contributes nothing to the existing body of knowledge. What the authors claim has an effect on GEOSAR imaging is well-known, and numerous papers have been published on the subject. Aside from that, the paper's writing style, terminology, methodology, solution, problems, pros and cons, and approach are all completely basic level of SAR information with no novelties. Hence, I would reject it and I will not recommend a resubmission.
For instance, authors can find some of the comments below, but the manuscript as a whole contains severe faults, and the comments are not limited to these.
1- Title: The title is neither accurate nor informative. The title is difficult to understand because of nonstandard terminology and poor English. The novelty must be explicitly stated in the title.
2- Abstract: Unfortunately, the abstract is poorly written, with bad English and nonstandard remote sensing taxonomy. In a nutshell, the abstract lacks a problem description, a solution, novelties, or verification metrics. It's worth mentioning that the use of nonstandard remote sensing terminology makes understanding the abstract difficult. Here just some comments about abstract:
2-1- Line 9: what is the difference between Doppler centroid and zero Doppler centroid?
2-2- Line 13: “During the aperture time” is correct, not the synthetic aperture
2-3- Line 14: what is the meaning of Doppler model? And what is the meaning of Doppler parameters while Doppler is a parameter per se!
2-4-Line 17: The authors claim that RLS variation has a significant impact on GEO-SAR Doppler performance, despite the fact that they declared just a few lines earlier in Line 13 that RLS and Doppler are significantly connected and can affect SAR imaging! In other words, what is the point of analyzing something that is already known and obvious? Is there anything new or novel you've added to the body of knowledge that you'd like to share?
2-5- Line 18: what is the meaning of stable azimuth resolution? what is the meaning of Doppler performance?
2-6- Line 17-19: It's not at all clear! What is the relationship between RLS and Doppler performance in real-time zero-Doppler control as compared to azimuth performance!! These specific lines are completely unacceptable.
3-Keywords: Keywords are not proper. Please use at least five keywords from your paper based on remote sensing taxonomy.
4- Introduction: Unfortunately, the introduction is quite poor. The problems have not been discussed at all, and the methodology is completely unknown. The approach toward the solution is not known, and the benefits are not clear. Citations are not good and contribution is not clear. In short, the Introduction is similar to simple sentences for Doppler estimation!
4-1- Please avoid using names! Just cite to the references in their proper lines.
4-2- references are mostly old and do not have anything to do with the topics. Please use highly accredited references close to the topic. For instance, [1-8] could have been summarized in just one or two references.
4-3- what is the difference between “pitch-yaw steering” and “yaw-pitch steering”. please be specific and write professionally.
4-4- Line 37-42: These lines are very important but the authors just summarized them in non-useful sentences. if you did something specific to this problem please describe it in your future work.
4-5- Line 46-48: Unfortunately, none of these statements is correct! There are many papers that go into great detail about the geometry configuration and parameter estimations. For instance, you can read “digital processing of synthetic aperture radar data by Ian. Cumming” and obtain a wealth of information on the subject!!
(To be continued with more serious problems)
Stay safe,
Round 2
Reviewer 1 Report
they corrected as requested.Reviewer 2 Report
The authors of this paper have made a satisfactory effort to improve the quality of the paper in accordance with the comments of the reviewers
Reviewer 3 Report
Despite the corrections, the new version was unable to convince me that the paper presents a compelling contribution to the body of knowledge. The simulation results are basic and have nothing to do with enhancing SAR signal processing. The paper's conclusion has already been published, and it has been applied to antenna steering mechanisms and Doppler centroid extraction, even with better formulation and justifications. Hence, i would reject it and do not encourage to resubmit it. authors must start from scratch.
Stay safe,